# LncRNA NEAT1/miR-146a-5p Axis Restores Normal Angiogenesis in Diabetic Foot Ulcers by Targeting mafG

**DOI:** 10.3390/cells13050456

**Published:** 2024-03-05

**Authors:** TCA Architha, George Raj Juanitaa, Ramanarayanan Vijayalalitha, Ravichandran Jayasuriya, Gopinathan Athira, Ramachandran Balamurugan, Kumar Ganesan, Kunka Mohanram Ramkumar

**Affiliations:** 1Department of Biotechnology, School of Bioengineering, SRM Institute of Science and Technology, Kattankulathur 603 203, Chengalpattu Dt., Tamil Nadu, India; tz8528@srmist.edu.in (T.A.); jg6002@srmist.edu.in (G.R.J.); vr3651@srmist.edu.in (R.V.); or jayasuriya199@gmail.com (R.J.); 2SRM Medical Hospital and Research Centre, SRM Institute of Science and Technology, Kattankulathur 603 203, Chengalpattu Dt., Tamil Nadu, India; athirag@srmist.edu.in (G.A.); balamurr@srmist.edu.in (R.B.); 3School of Chinese Medicine, Li Ka Shing Faculty of Medicine, The University of Hong Kong, 10 Sassoon Road, Pokfulam, Hong Kong 999077, China

**Keywords:** diabetic foot ulcers, lncRNA miRNA, miR-146a-5p/mafG axis, in silico analysis, angiogenesis

## Abstract

Non-healing lesions in diabetic foot ulcers are a significant effect of poor angiogenesis. Epigenetic regulators, mainly lncRNA and miRNA, are recognized for their important roles in disease progression. We deciphered the regulation of lncRNA NEAT1 through the miR-146a-5p/mafG axis in the progression of DFU. A lowered expression of lncRNA NEAT1 was associated with dysregulated angiogenesis through the reduced expression of *mafG*, *SDF-1α*, and *VEGF* in chronic ulcer subjects compared to acute DFU. This was validated by silencing NEAT1 by SiRNA in the endothelial cells which resulted in the transcriptional repression of target genes. Our in silico analysis identified miR-146a-5p as a potential target of lncRNA NEAT1. Further, silencing NEAT1 led to an increase in the levels of miR-146a-5p in chronic DFU subjects. This research presents the role of the lncRNA NEAT1/miR-146a-5p/mafG axis in enhancing angiogenesis in DFU.

## 1. Introduction

Diabetic foot ulcer (DFU) is a prevalent and serious consequence of type 2 diabetes, frequently leading to leg amputations [1,2]. Impaired angiogenesis is characterized by poor blood flow to the injured tissue, which contributes to the progression of DFU [1]. Perturbation to balance in the levels of pro-angiogenic and anti-angiogenic factors is closely associated with the development of diabetic foot ulcers [3]. A reduction in the levels of nitric oxide (NO) disturbs the redox homeostasis, causing oxidative stress, thus leading to endothelial dysfunction [4,5]. Recent reports have highlighted that targeting angiogenesis is a potentially promising strategy for the accelerated healing of diabetic ulcer wounds [6,7,8].

Endothelial cells crucial for angiogenesis exhibit a pro-invasive tendency and react to different stimuli [9]. Vascular endothelial growth factor (VEGF) and stromal cell-derived factor (SDF)-1α are key proteins that regulate angiogenesis and wound healing [10,11]. While VEGF promotes vessel permeability and the proliferation, migration, and survival of endothelial cells [11], SDF plays a major role in the migration and retention of endothelial cells at the sites of injury, thereby promoting neovascularization [10]. Further, epigenetic regulators are also suggested to be involved in regulating angiogenesis [12]. The long non-coding RNAs (lncRNAs) have been proven to affect angiogenesis by regulating angiogenic factors [13]. They influence the expression of several genes by acting as decoys, scaffolds, sponges, guides, and precursors to several factors [13,14]. Nuclear enriched abundant transcript 1 (NEAT1) is one such lncRNA that was noted as an essential structural component of the nuclear paraspeckles with five splice variants [15]. It is said to be a regulator of multiple genes and pathways, and reported for its role in cancer cell migration and metastasis [15,16,17,18].

mafG, a transcription factor is involved with pathways connecting glucose metabolism, antioxidant pathways, and bile homeostasis [19]. A recent study has documented that mafG regulates hepatic glucose metabolism through the lncRNA axis [20]. Several lncRNAs target the gene of interest directly or indirectly through miRNAs [21]. miRNAs interact with the 3’UTR of mRNAs, causing mRNA degradation and repressing translation, thereby controlling the regulation of mRNA [22]. Chen et al. found that the lncRNA NEAT1 enhances the course of non-alcoholic fatty liver disease by suppressing miR-146a, increasing ROCK1 expression, and aiding in the buildup of lipids in the liver [23]. Analyzed microRNA target detection revealed a specific binding site between lncRNA NEAT1 and miR-146a-5p. Therefore, we propose that the long non-coding RNA NEAT1 could impact angiogenesis by interacting with miR-146a-5p. This work aims to investigate how the relationship between lncRNA NEAT1 and miR-146a-5p affects the course of DFU by influencing the activities of endothelial cells.

## 2. Materials and Methods

### 2.1. Enrolment of the Study Subjects

Tissue biopsies were collected from a total of 60 DFU subjects (27 acute and 33 chronic) who had diabetes between 6 and 15 years and no complications other than DFU. Each participant provided informed consent after being briefed on the study in their local language. A qualified podiatrist at SRM Medical Hospital and Research Center categorized the grades of DFU according to IDSA-IWGDF recommendations. All tissues were obtained with institutional ethical approval (1901/IEC/2020). Superficial ulcers without infection (grade-1) were classified as acute DFU, while ulcers that had penetrated the skin or reached the bone with infection (grade-2, -3, and -4) were identified as chronic ulcers. The collected tissues (approx. 100 mg) were washed in PBS and immediately transferred to an RNA stabilization solution, and stored in a deep freezer until further use.

### 2.2. Cell Culture Treatments and Conditions

The human endothelial cell line (EA.hy926) was grown in DMEM (Sigma, St. Louis, MO, USA; Burlington, MA, USA) supplemented with 10% FBS (Gibco, Grand Island, NY, USA) at 37 °C in a humidified incubator with 5% CO_2_. The cells were maintained in DMEM with 1% FBS for 3 h before any treatment. The cells were incubated with extra glucose (33.3 mM) [24] and a cytokine cocktail containing IL-1β (Abcam, Cambridge, MA, USA), TNF-α (R&D systems, Minneapolis, MN, USA), and IFN-ϒ (Abcam, USA) at 10 ng/mL concentrations each for 24 h to induce hyperglycemic stress. The cells were trypsinized and stored in RNA stabilization at −80 °C until further use.

### 2.3. RNA Isolation

About 50 mg of frozen tissue was weighed and ground with liquid nitrogen using a pre-chilled mortar and pestle inside a biosafety cabinet to maintain aseptic conditions. RNaze*zap* RNase Decontamination Solution (Invitrogen, Thermo Fisher Scientific, Waltham, MA, USA) was used to decontaminate the surface, and RNaseOUT Recombinant Ribonuclease Inhibitor (Invitrogen) was added to the sample to inhibit the activity of RNases. About 500 μL QIAzol RNA isolation reagent (Qiagen, Germantown, MD, USA) was added to the ground tissues and mixed, followed by 200 μL of chloroform. Similarly, the harvested cells were lysed using 300 μL QIAzol and added with 200 μL of chloroform. The supernatant after centrifugation was isolated and mixed with an equal proportion of 70% ethanol (Hayman, Witham, UK). The contents were moved to a spin column included in the RNeasy kit from Qiagen and RNA isolation was performed following the manufacturer’s instructions. Finally, two fractions enriched with large and small RNA were eluted separately.

### 2.4. cDNA Synthesis and q-RT-PCR

The purity of both the large and small RNA-enriched fractions were quantified on a nanodrop spectrometer, and further taken for cDNA conversion. The mature miRNA was selected and converted from a small RNA-enriched fraction using HiSpec buffer provided with the miScript II RT kit (Qiagen, USA). For lncRNA-cDNA, first-strand synthesis was carried out from a large RNA-enriched fraction using an RT2 first-strand kit (Qiagen, USA). The mRNA was converted to cDNA from a large RNA fraction using an oligo dT mixture provided with the kit (Bio-Rad, Hercules, CA, USA). To analyze the expression of the targets of the study, we used the respective primers as listed in Table 1. The lncRNA was quantitatively determined using RT2 SYBR green master mix (Qiagen, USA), and SYBR green super mix (Bio-Rad) was used to detect the target mRNAs. GAPDH served as an internal control for both lncRNA and mRNAs. With respect to miRNAs, miScript SYBR Green (Qiagen) was used with U6 as an internal control.

### 2.5. Protein Extraction and Multiplexing of Angiogenic Markers from the Tissue Biopsies of Study Subjects

Ten tissue biopsies were chosen at random from acute and chronic ulcer patients and subjected to total protein extraction using the ProteoExtract Complete Mammalian Proteome Extraction Kit (MilliporeSigma, Burlington, MA, USA). The frozen tissue was ground using liquid nitrogen, transferred to a pre-chilled Eppendorf, and immediately subjected to total protein extraction. The supernatant of each sample was collected in a fresh microcentrifuge tube, and the concentration of total protein was quantified using Bradford’s protein assay reagent. The Bio-Plex Pro Human Cytokine multiplex assay (Bio-Rad) was used to quantify the levels of angiogenic markers in the tissue supernatants using a Bio-Plex 200 Multiplex System. A normalized concentration of protein was loaded to the wells and incubated with antibody as per the manufacturer’s experimental design. The concentration of each analyte measured using multiplexing was then normalized to the total protein concentration of that particular sample.

### 2.6. Silencing lncRNA NEAT1 and miR-146a-5p in Endothelial Cells

Antisense oligonucleotides targeting lncRNA NEAT1 and miR-146a-5p along with an off-target scrambled control were synthesized (listed in Table 2). The oligonucleotides were transiently transfected into 60–70% confluent cells using lipofectamine 2000 transfection reagent (Invitrogen). The transfection complex was replenished with medium after 4 h.

### 2.7. Scratch Assay

An in vitro scratch assay was conducted to verify the impact of lncRNA NEAT1 on the proliferation and migration of endothelial cells. The cells were transfected with antisense oligonucleotides targeting Si-NEAT1 and SC in a 6-well plate, as described earlier. Post recovery, the cells were cultured in new DMEM supplemented with 10% FBS. A scratch was created in the 6-well plate using a sterile 10 µL tip and observed for cell movement. The photos were captured 8 h later using an inverted microscope.

### 2.8. In Vitro Tube-Formation Assay

A tube-formation assay was performed to assess the angiogenic ability of endothelial cells after silencing lncRNA NEAT1. The endothelial cells were transfected with Si-NEAT1 and SC as mentioned in the previous sections. After recovery, the cells were seeded on a Matrigel basement membrane extract (Corning, NY, USA) in fresh DMEM with 10% FBS. After 8 h, the cells were observed under microscope for network formation. The cells were imaged using an inverted microscope.

### 2.9. Establishment of lncRNA-miRNA-mRNA Network

The list of miRNAs that regulate *mafG* was retrieved from the miRwalk database (http://mirwalk.umm.uni-heidelberg.de/, accessed on 15 November 2023). Pre-validated miRNAs were excluded from the list and miRNAs binding to 3′ UTR were short-listed. After this, the miRNAs with at least 1 seed region and binding gap were selected. The selected miRNAs were further analyzed based on the number of pairings and largest consecutive pairing. The resulting miRNAs were loaded onto Venny v2.1 (https://bioinfogp.cnb.csic.es/tools/venny/, accessed on 15 November 2023) along with the list of miRNAs regulated by lncRNA NEAT1 acquired from the starBase database v2.0 (http://starbase.sysu.edu.cn/, accessed on 15 November 2023). The common miRNAs were further analyzed based on binding energy and a heat map was generated, based on which miR-146a-5p was selected as a potential target in this study.

### 2.10. Statistical Analysis

One-way ANOVA was utilized to compare the average variable among groups in the biochemical parameter table. The expression data between the groups were analyzed using a Student’s *t*-test. The analyses were conducted using GraphPad Prism software version 8.0. Pearson’s correlation was utilized to assess the correlation among the study markers. Significance was determined for all studies at a *p*-value of less than 0.05.

## 3. Results

### 3.1. Clinical and Biochemical Factors of the Study Participants

The wound size was measured by metric analysis and found to be big in chronic DFU patients compared with the acute ones (Figure 1a). Along with this, two major clinical parameters that reveal the inflammatory status in the body such as C-reactive protein (CRP) and white blood cell (WBC) count also increased in patients with chronic DFU (Figure 1b,c). The biochemical parameters of the study subjects are represented in Table 3. Also, glycated haemoglobin (HbA1c) and low-density lipoprotein cholesterol (LDL-c) were significantly high (*p* < 0.05) in patients with chronic ulcers compared to patients with acute ulcers.

### 3.2. Levels of Angiogenic Markers among the Study Subjects

Angiogenesis play a vital role in wound healing, and the markers VEGF and SDF-1α are known to promote angiogenesis. The concentrations of angiogenic markers were quantified by multiplexing. As seen in Figure 2a, the concentration of SDF-1α in acute DFU patients was 62.8 ± 14.1, which seemed to be significantly reduced (*p* < 0.001) to about 22.4 ± 9.5 in patients with chronic DFU. Similarly, the concentration of VEGF (Figure 2b) in acute DFU patients was 352.9 ± 57.1, which seemed to be significantly reduced (*p* < 0.001) to about 161.9 ± 41.6 in patients with chronic DFU.

### 3.3. Expression of Angiogenic Markers and lncRNA NEAT1 among the Study Subjects

This study aimed to understand the regulatory mechanism of angiogenesis through non-coding RNAs. We initially measured the mRNA levels of angiogenic factors *mafG*, *SDF-1α* and *VEGF*, as well as the expression of lncRNA NEAT1 in the participants of the study. Figure 3a displays a notable reduction of angiogenic markers including *mafG* (*p* < 0.001), *SDF-1α* (*p* < 0.001), and *VEGF* (*p* < 0.001) in individuals with chronic DFU. The expression of lncRNA NEAT1 was decreased by approximately 2.5-fold (*p* < 0.001) in chronic DFU cases compared to acute DFU cases (Figure 3b).

### 3.4. Inhibition of lncRNA NEAT1 Reduced Angiogenesis and Migration in Endothelial Cells In Vitro

We tested the human endothelial cells with an HGM to confirm the results obtained from clinical samples. Angiogenic indicators *VEGF* and *SDF-1α* did not show any significant changes initially (Appendix A), and subsequently reduced at 24 h. Consistent with our clinical results, a reduced production of angiogenic markers including *mafG* (3-fold; *p* < 0.001), *SDF-1α* (2.3-fold; *p* < 0.01), and *VEGF* (2.8-fold; *p* < 0.001) was noted in endothelial cells generated by HGM compared to the control group (Figure 4a). A 4-fold decrease (*p* < 0.01) was detected in the expression of lncRNA NEAT1 in endothelial cells generated by HGM compared to the control group (Figure 4b).

To clarify the specific function of lncRNA NEAT1 in regulating angiogenesis, we suppressed its expression using an antisense oligonucleotide (Si-NEAT1) and compared it with a scramble control (SC). Figure 4c shows a notable 3-fold drop in lncRNA NEAT1 expression (*p* < 0.01) in Si-NEAT1. The association of lncRNA NEAT1 in angiogenesis was confirmed by significantly reduced expressions of markers such as *mafG* (2.5-fold, *p* < 0.01), *SDF-1α* (2-fold, *p* < 0.05), and *VEGF* (2.5-fold, *p* < 0.01) in Si-NEAT1 (Figure 4d). Further, endothelial cell migration is one of the essential parameters for the regulation of angiogenesis. From Figure 5a, we observed almost 2-fold decreased cell migration (*p* < 0.05) on inhibition of lncRNA NEAT1 compared to SC. Moreover, the angiogenic ability of NEAT1-silenced endothelial cells was assessed using an in vitro tube-formation assay. As seen in Figure 5b, we observed a significant decrease in the tube-formation ability of NEAT1-silenced endothelial cells, compared to SC.

### 3.5. miR-146a-5p as a Putative Target for lncRNA NEAT1

lncRNAs act as miRNA sponges to control the action of miRNA, reducing their effect on target mRNAs. Hence, to select the putative target of lncRNA NEAT1, we adapted an in silico screening. Figure 6a represents the overall workflow followed to determine the potential miRNA. The common miRNAs that are regulated by lncRNA NEAT1 and that regulate mafG were screened (Figure 6b). The binding score of targets was evaluated for all selected miRNAs and that of miR-146a-5p was observed to be the highest (Figure 6c). Figure 6d represents the targeted binding site of miR-146a-5p and lncRNA NEAT1. To determine miR-146a-5p as a putative target of lncRNA NEAT1, the expression of target miRNA was assessed in Si-NEAT1 and was found to be increased (1.5-fold, *p* < 0.05), compared to SC (Figure 6e).

### 3.6. Correlation Analysis of lncRNA-miRNA-mRNA Axis

Based on the observations in Section 3.5, the expression of miR-146a-5p was assessed in DFU subjects. An almost 3-fold (*p* < 0.001) increased expression of miR-146a-5p was seen in chronic DFU compared to that of acute DFU (Figure 7a). To determine the association of target lncRNA and miRNA with the study markers, Pearson’s correlation analysis was performed. Figure 7b shows a significant negative connection between miR-146a-5p and lncRNA NEAT1, as well as other angiogenic targets. Further, we also evaluated the clinical evidence in vitro using endothelial cells exposed to a hyperglycemic microenvironment (Figure 7c), and found upregulated miR-146a-5p expression (1.5-fold; *p* < 0.01), which negatively correlated with lncRNA NEAT1, and angiogenic markers other than SDF-1α (Figure 7d). From the correlation analysis, it was also observed that lncRNA NEAT1 has a strong positive correlation with mafG in both chronic DFU subjects (*p* < 0.001) and HGM-induced endothelial cells (*p* < 0.01).

### 3.7. mafG as a Putative Target for miR-146a-5p

As mentioned above, miRNAs indulge in mRNA repression by binding to their 3′UTR region. To elucidate the action of selected miRNA on mafG, we predicted the binding of miR-146a-5p to the 3′UTR of mafG. As there was a negative correlation between miR-146a-5p and mafG in both chronic DFU subjects (*p* < 0.001) and HGM-induced endothelial cells (*p* < 0.05), as mentioned in Section 3.6, we examined the association of miR-146a-5p on the regulation of angiogenesis. The targeted binding site of *mafG* with miR-146a-5p is represented in Figure 8a. The target *mafG* was confirmed as the putative target for miR-146a-5p using antisense oligonucleotides that target miR-146a-5p (Si-miR146a). As represented in Figure 8b, the expression of miR-146a-5p was observed to be reduced by 2.2-fold (*p* < 0.01), whereas the expression levels of both mafG and VEGF were increased by 1.5-fold (*p* < 0.05) (Figure 8c). This confirms the mafG as a putative target for miR-146a-5p, and a decreased *mafG* suppresses angiogenesis through the downregulation of *VEGF*.

## 4. Discussion

Diabetic foot is characterized by improper peripheral blood flow, contributing to poor cell growth and the release of vascular growth factors, which altogether delays healing in DFU patients [25]. Biochemical parameters such as LDL-c and HbA1c showed a significant difference between the study groups. Previously, we have reported the differences in several parameters among healthy volunteers compared with type 2 diabetes and DFU subjects [26,27]. CRP and WBC were increased in patients with chronic ulcers compared to acute DFU subjects. This corresponded to the larger wound size noted in chronic DFU patients. Several intrinsic factors are plausibly connected to the poor closure of wounds, of which one critical factor is dysregulated angiogenesis [28,29].

Onodera et al. reported multiple organ failure in mice deficient with mafG and mafK, proving lethal [30]. The mafs were identified as chief regulators of angiogenesis through *VEGF* transcriptional response by Wang et al. in 2019 [31]. In the present study, we observed decreased protein levels of both angiogenic markers on chronic ulcer patients compared to acute ulcer patients, which coincided with their mRNA levels. The result was validated in vitro using HGM-induced endothelial cells, with a reduced expression of angiogenic targets compared to the control.

NEAT1 is an abundantly expressed lncRNA widely studied in cancer progression, tumor cell migration, invasion, metastasis, and epithelial-to-mesenchymal transition [15]. We observed a decreased expression of lncRNA NEAT1 in chronic DFU subjects compared to acute DFU, and validated the same in HGM-induced endothelial cells in vitro, suggesting a positive correlation. Shoa et al. elucidated the reduction in the expression of *VEGF* and *TGF-β1* following the silencing of lncRNA NEAT1 in retinal endothelial cells [32]. The lncRNA NEAT1 was knocked down using antisense oligonucleotides in endothelial cells to confirm the association of lncRNA NEAT1 with *mafG*. The inhibition of lncRNA NEAT1 suppressed *mafG* and other angiogenic markers in endothelial cells. These findings show that lncRNA NEAT1 regulates *mafG* and might be essential in regulating angiogenesis in DFU. Along with this, we observed a reduced cell migration and tube formation on knockdown of lncRNA NEAT1, emphasizing its role in migration and proliferation.

The dysregulation of non-coding RNAs is reportedly involved in the pathogenesis of various diseases, including DFU [33,34,35]. Competitive endogenous RNA (ceRNA) binding is a novel hypothesis where the endogenous miRNA shares its miRNA response element (MRE) with lncRNA [36,37]. Thus, the lncRNA competitively binds to the MREs and sponges the target miRNA, inhibiting the action of a particular miRNA [36]. For example, Yuan et al. demonstrated the sponging of miR-133a by lncRNA NEAT1 in mice cervical carcinoma. *Sox4* was identified as a putative target of miR-133a and was released by the action of lncRNA, with the researchers concluding that the enhanced level of Sox4 led to the progression of cervical carcinoma in mice [38]. Similarly, our study found miR-146a-5p as a putative target of lncRNA NEAT1 through in silico analysis. Further, *mafG* was also identified as a potential target for miR-146a-5p. These results were validated in vitro, proving that miR-146a-5p is indeed a target of lncRNA NEAT1.

The levels of miR-146a-5p were high in chronic DFU subjects compared to their acute counterparts, and in endothelial cells exposed to HGM in vitro compared to the control. This finding is in line with other reports where higher levels of miR-146a were reported in DFU and atherosclerosis cases [39,40,41]. Contradictory to our data, a lowered expression of miR-146a was assessed which was seen to delay healing response in diabetic mice [42]; however, few studies have suggested miR-146a as inflammation-inducible miRNA in various disease conditions [43,44,45]. miR-146a is most commonly known for its role in inflammation-related diseases [46]. It is found to impede the immune response by playing an agonist role in inflammatory conditions [47]. The miRNA-146a was identified to regulate NF-kB and modulate migration and apoptosis in vascular smooth cells [48]. Also, miR-146a was identified as a biomarker in patients with cardiovascular disease. Higher levels of miR-146a were reported to significantly correlate with the severity of the disease [49]. The predicted binding site for miR-146a-5p and *mafG* in silico was validated in endothelial cells by silencing miR-146a-5p using siRNA. The miR-146a-5p inhibition was confirmed by observing a decrease in miR-146a in Si-miR146a compared to SC. We also observed an increase in the levels of *mafG* and *VEGF* in Si-miR146 compared to SC. Thus, these findings indicate that miR-146a-5p is a putative target of lncRNA NEAT1, and that miR-146a-5p drives angiogenesis through mafG.

## 5. Conclusions

Our research has clarified the function of the lncRNA NEAT1/miR-146a-5p/mafG axis in controlling angiogenesis in DFU. The downregulation of lncRNA NEAT1 caused elevated levels of miR-146a-5p, which inhibited mafG expression and impaired angiogenesis in DFU.

## Figures and Tables

**Figure 1 cells-13-00456-f001:**
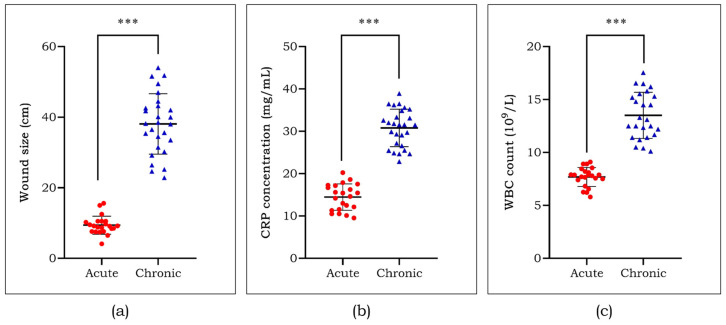
Assessment of wound size (**a**), C-reactive protein (**b**), and white blood cell count (**c**) among the study groups. Data are represented as mean ± S.D. *** *p* < 0.001.

**Figure 2 cells-13-00456-f002:**
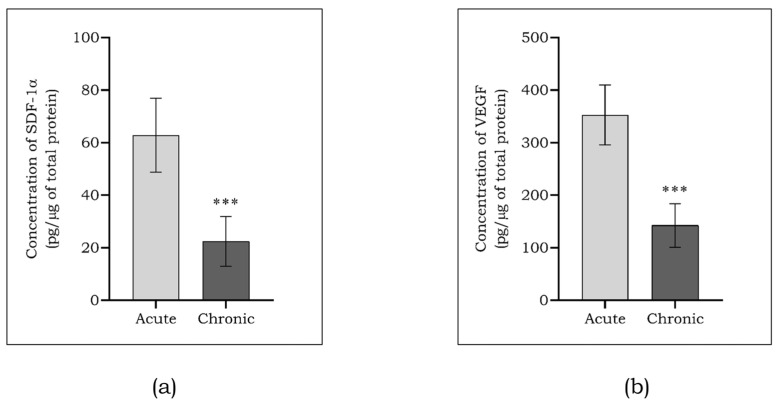
Levels of angiogenic markers such as SDF-1α (**a**) and VEGF (**b**) among the study subjects assessed using a Multiplex assay. Data are represented as mean ± S.D. *** *p* < 0.001.

**Figure 3 cells-13-00456-f003:**
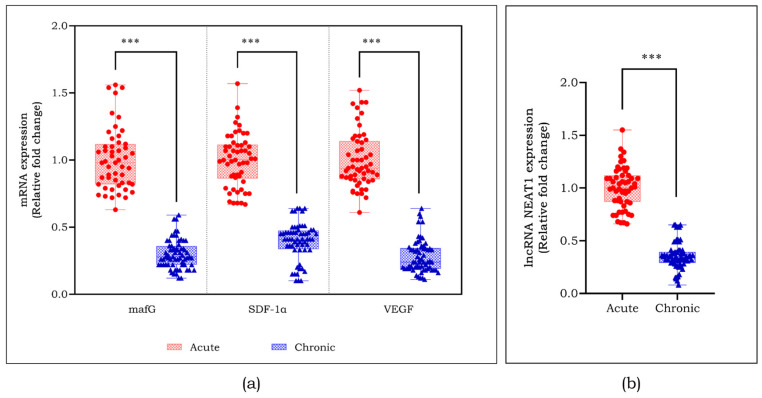
mRNA levels of angiogenic factors such as (**a**) *mafG*, *SDF-1α*, and *VEGF*, and (**b**) levels of lncRNA NEAT1 among the study participants assessed using q-RT-PCR. Data are represented as mean ± S.D. *** *p* < 0.001.

**Figure 4 cells-13-00456-f004:**
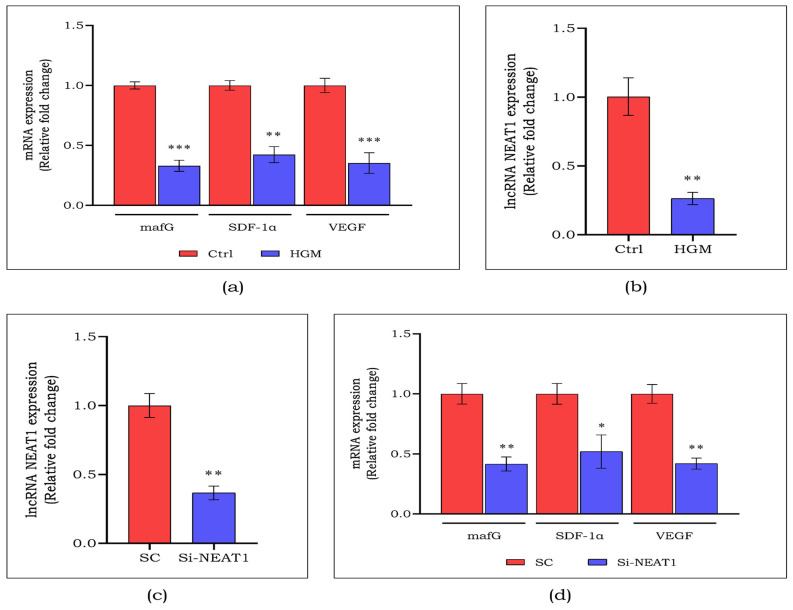
Expression of angiogenic targets (**a**) such as *mafG*, *SDF-1α*, *VEGF*, and lncRNA NEAT1 (**b**) in HGM-induced endothelial cells. Expression of lncRNA NEAT1 (**c**) and angiogenic targets (**d**) on siRNA-mediated silencing of lncRNA NEAT1. Data are represented as mean ± S.D.* *p* < 0.05, ** *p* < 0.01, *** *p* < 0.001.

**Figure 5 cells-13-00456-f005:**
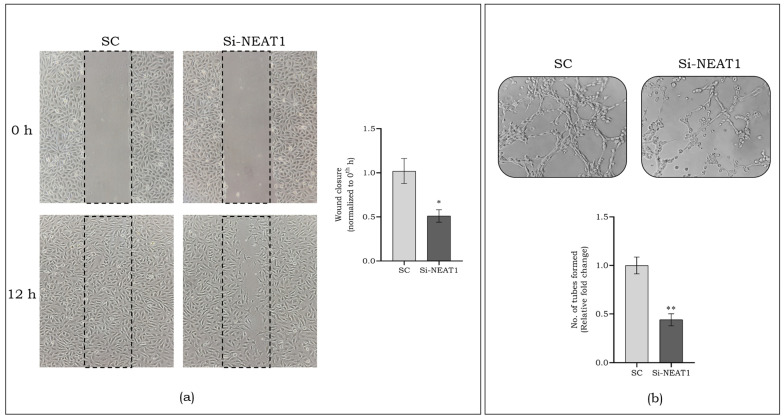
Estimation of endothelial cell migration on silencing lncRNA NEAT1 (**a**), Magnification 10×. Tube formation ability of endothelial cells on silencing lncRNA NEAT1 (**b**), Magnification 10×. Data are represented as mean ± S.D. * *p* < 0.05, ** *p* < 0.01.

**Figure 6 cells-13-00456-f006:**
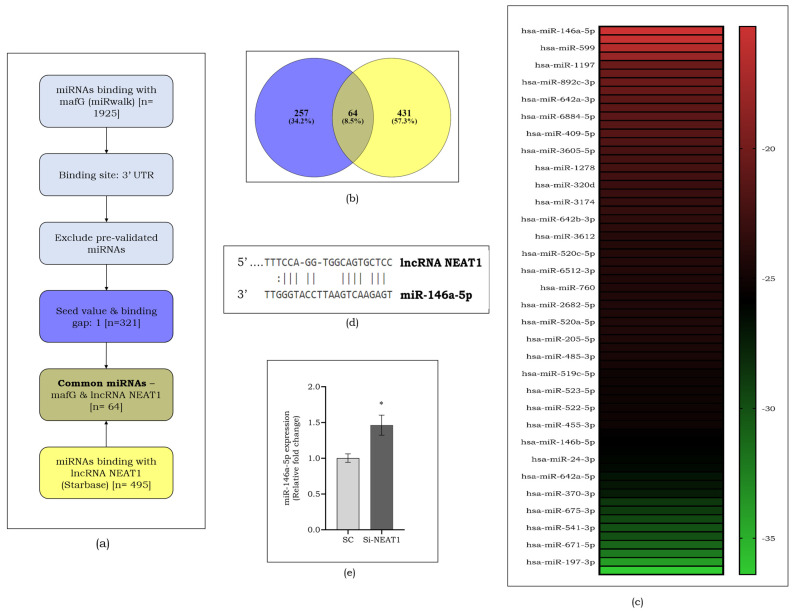
Workflow for selection of miRNAs that regulate mafG and that are regulated by lncRNA NEAT1 obtained from the StarBase and miRwalk online databases (**a**). Venn diagram showing common miRNAs between mafG and lncRNA NEAT1 (**b**). Heat map showing the list of selected miRNAs and their binding energy (**c**). Targeted binding site of lncRNA NEAT1 and miR-146a-5p (**d**). Expression of miR-146a-5p on NEAT1-silenced endothelial cells (**e**). Data are represented as mean ± S.D. * *p* < 0.05.

**Figure 7 cells-13-00456-f007:**
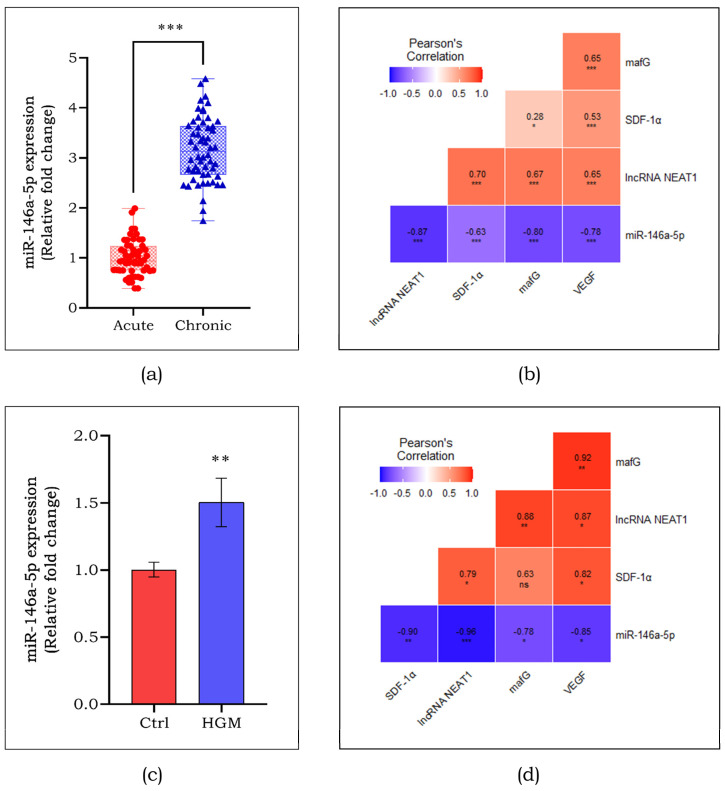
Expression of miR-146a-5p among acute and chronic DFU subjects (**a**). Pearson’s correlation analysis for the study markers among acute and chronic DFU subjects (**b**). Expression of miR-146a-5p in HGM-induced endothelial cells (**c**). Pearson’s correlation analysis for the study markers in HGM-induced endothelial cells (**d**). Data are represented as mean ± S.D. * *p* < 0.05, ** *p* < 0.01, *** *p* < 0.001.

**Figure 8 cells-13-00456-f008:**
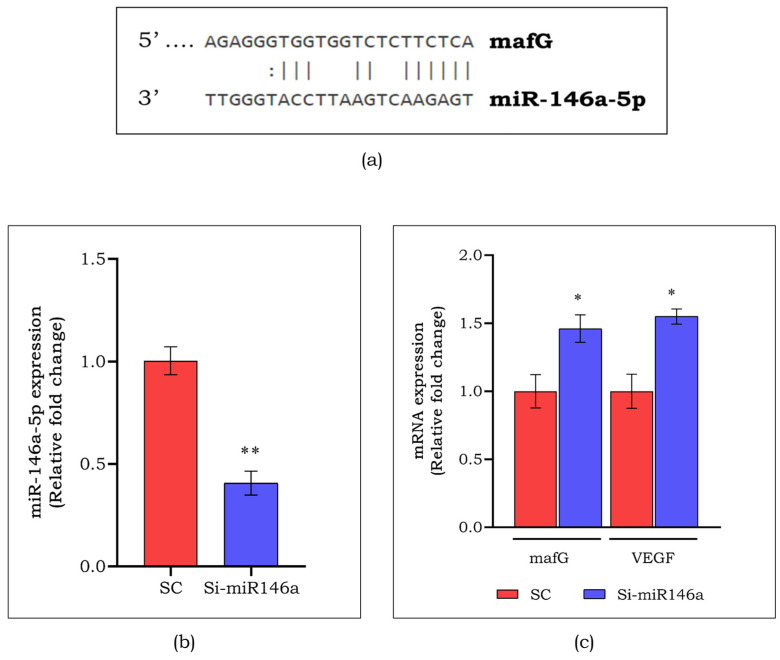
Targeted binding site of miR-146a-5p and *mafG* (**a**). Expression of miR-146a-5p (**b**) and angiogenic targets *mafG* and *VEGF* (**c**) upon silencing miR-146a-5p in endothelial cells. Data are represented as mean ± S.D. * *p* < 0.05, ** *p* < 0.01.

**Table 1 cells-13-00456-t001:** Primer list for this study.

Target Name	Primer Sequence	
	Forward	Reverse
*mafG*	CTGTTTTCCCGTGTTCGTTT	ACCCCAGTTTCACCTACCCC
*SDF-1α*	CGCACTTTCACTCTCCGTCA	AGCACGACCACGACCTTG
*VEGF*	CTACCTCCACCATGCCAAGT	GCAGTAGCTGCGCTGATAGA
lncRNA NEAT1	CTTCCTCCCTTTAACTTATCCATTCAC	CTCTTCCTCCACCATTACCAACAATAC
*GAPDH*	AAGAAGGTGGTGAAGCAGGC	GTCAAAGGTGGAGGAGTGGG
miR-146a-5p	TGAGAACTGAATTCCATGGGTT	
U6	CGCAAGGATGACACGCAAATTC	

**Table 2 cells-13-00456-t002:** Antisense oligonucleotide list for this study.

Antisense Oligonucleotides	Sequence
Si-NEAT1	CTGTTTTCCCGTGTTCGTTT
SC for NEAT1	CGCACTTTCACTCTCCGTCA
Si-miR146a-5p	CTACCTCCACCATGCCAAGT
SC	ACGTCTATACGCCCA

**Table 3 cells-13-00456-t003:** Clinical and biochemical factors of the participants involved in this study.

Clinical Parameters	Acute DFU (*n* = 27)	Chronic DFU (*n* = 33)
Age (Years)	48.8 ± 4.9	51.4 ± 2.6
BMI (kg/m^2^)	28.6 ± 1.6	29.2 ± 3.5
SBP (mm Hg)	135.2 ± 3.1	140.6 ± 6.2
DBP (mm Hg)	85.4 ± 3.1	89.5 ± 3.3
FPG (mg/dL)	186.8 ± 8.0	210.9 ± 10.3
PPG (mg/dL)	215.5 ± 6.0	275.5 ± 12.5
HbA1c (%)	7.2 ± 1.0	10.8 ± 1.6 *
TSC (mg/dL)	179.3 ± 4.9	191.8 ± 5.4
HDL-c (mg/dL)	47.5 ± 5.1	40.1 ± 4.6
LDL-c (mg/dL)	100.1 ± 7.4	135.5 ± 21.9 *
Urea (mg/dL)	30.2 ± 2.6	34.5 ± 3.1
Creatinine (mg/dL)	1.1 ± 0.1	1.12 ± 0.1

Data presented in the table is represented as mean ± S.D. * *p* < 0.05.

## Data Availability

The datasets generated during and/or analyzed during the current study are available from the corresponding author on reasonable request.

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
