# Peer review of "LncRNA NEAT1/miR-146a-5p Axis Restores Normal Angiogenesis in Diabetic Foot Ulcers by Targeting mafG"

_cells, 2024, doi:10.3390/cells13050456_

Round 1

Reviewer 1 Report

Comments and Suggestions for Authors

The authors analysed the expression of mafG, SDF-1α and VEGF angiogenic markers and correlation with the expression of lncRNA NEAT1in tissue biopsies of 50 DFU subjects (27 acute and 33 66 chronic). In chronic DFU cases angiogenic markers and lncRNA NEAT1 seemed to be significantly reduced if compared to acute DFU cases. The human endothelial cells (EA.hy926) were exposed to HGM and used to validate the data observed in DFU patients. At the end they found that miR-146a-5p is a putative target for lncRNA NEAT1 and that mafG is a target of miR-146a-5p, suggesting that lncRNA NEAT1/miR-146a-5p/mafG axis improve angiogenesis in DFU patients. The finding that miR-133a are sponged by lncRNA NEAT1 is not necessarily novel, but was never shown in endothelia cells and linked to chronic DFU. An overall good paper, the methods used in the study and the results are well described and discussed.

Just a few comments regarding some of the experiments.

Comments:

1- in line 185, 188 and in fig. 2 legend the concentration” should be “the relative expression”;

2- figure 6C is quite impossible to read, resize the figure or put some data in the supplemental;

3- in the conclusion the statement “We have thus demonstrated the function” should be more mild since was only demonstrated a strong correlation that link lncRNA NEAT1/miR-146a-5p/mafG gene, no interaction studies are presented to validate the functional axis.

Reviewer 2 Report

Comments and Suggestions for Authors

In the present manuscript, Architha et al. demonstrate the importance of lncRNA NEAT1/miR-146a-5p axis in restoring angiogenesis via targeting mafG in DFU. Although, the concept is novel and the study is interesting, there are some concerns which prevent accepting the manuscript in the present form and are needed to be taken care of. The comments are as follows:

1. Thorough revision of the English language is required. For example, abstract line 2: should be 'considered as a major consequence....'; line 3: should be 'recognized for their roles in disease.....' and many more throughout the text.

2. It is recommended to include the full name of abbreviations were used for the first time.

3. The title of main manuscript is different than the title of the supplementary. Please select anyone.

4. In results section, in each subsection a brief description of experimental approach is required.

5. Controls are missing in the following figures: Fig. 4c, 4d, 5, 6e, 8b, and 8c; Control without transfection (either scrambled or specific siRNA) should be provided to see the effect of transfection.

6. In Fig. 4c, lnc NEAT1 expression is shown to be reduced more than 2.5-fold upon siRNA transfection whereas in Fig. 5 the migration is retarded by only 1.5-fold. Please explain this discrepancy.

7. Page 8, line 240: Is it mafG or it would be lncRNA NEAT1? Please check.

8. Please increase the font size of labeling of Fig. 6c- heatmap as hardly any miRNA is visible.

9. It would be nice to show the overexpressing effect of miR146a-5p by miR146a-5p mimic transfection in endothelial cells- on mafG, VEGF expression and associated migration.

10. Endothelial migration upon miR146a-5p inhibition as well as overexpression should be provided.

11. The authors could simply check the angiogenic potential by in vitro angiogenesis assay on matrigel upon NEAT1, miR146a-5p knockdown and overexpression in endothelial cells.   

Comments on the Quality of English Language

The study is novel and interesting but requires significant revision before acceptance. I would recommend sending it to a major revision.

Reviewer 3 Report

Comments and Suggestions for Authors

The paper of Dr. Architha, Dr. Kumar Ganesan, Dr. Kunka Mohanram Ramkumar, and co-authors is presenting proofs of the involvement of LncRNA NEAT1/miR-146a-5p in angiogenesis of Diabetic foot ulcers through mafG signaling. The paper is interesting and presents proofs on biopsies along with in vitro demonstration. However, some methodological details have to be clarified before publication. I propose major modifications and I have to underline that the second major comment needs additional experiments to be addressed properly.

Major comments

(1) The working plan is unclear, even if you got official clearance for the clinical trial. You had biopsies from 27 acute and 33 chronic patients (2.1.). But the section 2.5. is stating that only ten biopsies of each group were chosen at random. Did you use a biopsy from 60 patients for q-PRC and only a biopsy from 20 patients for protein extraction? Thanks for altering accordingly.

(2) Only 2 reference genes have been used, gapdh (for lncRNA and mRNA) and u6 (for miRNA), in contradiction with international standards asking for 3 reference genes. At least a second reference gene for each category is needed to be considered for publication.

Minor comments :

(1) First author’s first name is needed in full not abbreviated (TCA Architha).

(2) You are stating at the end of introduction that « lncRNA NEAT1 and miR-146-5p might function in relay, and influence angiogenesis ». You conclusion is « repression of lncRNA NEAT1 induced miR-146a-5p resulting in mqfG inhibition ». This is more a network functionning in antagonism than in relay. This may be underlined in your conclusion.

Comments on the Quality of English Language

The English style is correct.

Round 2

Reviewer 2 Report

Comments and Suggestions for Authors

The authors tried to answer all my questions. I am satisfied with the answers.  

Author Response

We would like to express our sincere gratitude for your time and effort in reviewing our manuscript. Your insightful feedback and constructive comments have been invaluable in improving the quality and clarity of our work. Your expertise and insights are deeply appreciated.

Reviewer 3 Report

Comments and Suggestions for Authors

The paper of Dr. Architha, Dr. Kumar Ganesan, Dr. Kunka Mohanram Ramkumar, and co-authors is presenting proof of the involvement of LncRNA NEAT1/miR-146a-5p in angiogenesis of Diabetic foot ulcers through mafG signaling.

The addition of at least a second gene for normalization, or even better, a third is mandatory for publication.

So, I have to recommend your manuscript for major revision or rejection.

I understand that it is a costly major modification, but the international standard in qPCR has moved since 15-20 years in that direction.

Comments on the Quality of English Language

The English style is correct.

Author Response

Thank you for your thorough evaluation of our manuscript. We appreciate your insightful comments and the opportunity to address the concerns raised regarding the normalization approach in our qPCR analysis. We understand the importance of using multiple reference genes to enhance the accuracy and reliability of gene expression measurements, particularly in qPCR assays.

While we agree that including a third gene for normalization may offer additional stability and confidence in the normalization process, we followed a stringent experimental approach that included appropriate controls and validation techniques to ensure the validity and reliability of our results. In instances where data inconsistency or reliability is observed, we incorporate normalization using a third gene. Our research focus on investigating the involvement of LncRNA NEAT1/miR-146a-5p in angiogenesis of diabetic foot ulcers through mafG signaling. As such, our experimental design was tailored to assess the expression levels of these specific genes.

Adding a third reference gene for normalization may not always lead to better interpretation of results. In some cases, it may introduce complexity without providing substantial improvements in data quality or reliability, particularly if the two selected reference genes exhibit consistent and stable expression patterns.

Additionally, we appreciate your acknowledgment of the resource-intensive nature of conducting qPCR assays with multiple reference genes, which aligns with our considerations in the experimental design process.

Furthermore, we have extensive experience in studying the epigenetic regulation of target genes in diabetic complications, as evidenced by our previous publications in reputed journals. Our research outcome has been dedicated to elucidating the molecular mechanisms underlying diabetic foot ulcers and related complications, which has provided us with valuable insights into the regulatory pathways involved.

  • Jayasuriya et al. Role of Nrf2 in MALAT1/ HIF-1α loop on the regulation of angiogenesis in diabetic foot ulcer. Free Radic Biol Med. 2020 Aug 20;156:168-175. doi: 10.1016/j.freeradbiomed.2020.05.018. Epub 2020 May 27. PMID: 32473205.
  • Amin et al. MiR-23c regulates wound healing by targeting stromal cell-derived factor-1α (SDF-1α/CXCL12) among patients with diabetic foot ulcer. Microvasc Res. 2020 Jan;127:103924. doi: 10.1016/j.mvr.2019.103924. Epub 2019 Sep 11. PMID: 31520606.
  • Sakshi et al. MicroRNA-27b Impairs Nrf2-Mediated Angiogenesis in the Progression of Diabetic Foot Ulcer. J Clin Med. 2023 Jul 7;12(13):4551. doi: 10.3390/jcm12134551. PMID: 37445586; PMCID: PMC10342788.
  • Milan et al. MiR125b regulates Vitamin D resistance by targeting CYP24A1 in the progression of gestational diabetes mellitus. Journal of Steroid Biochemistry and Molecular Biology (2024: In press)

We assure you that we have carefully considered your recommendation, and we are open to addressing any further concerns or suggestions for improvement in our revision.

We sincerely hope that our manuscript will still be considered for publication in your esteemed journal.

Thank you once again for your time, consideration, and valuable feedback.